# Indoor methane consistently above outdoor levels in homes with natural gas service

Nathan G. Phillips[1]*, Robert Ackley[2], Andee Krasner[3]

**1** Department of Earth and Environment, Boston University, Boston, Massachusetts United States of America, **2** Gas Safety, Inc., Southborough, Massachusetts, United States of America, **3** Indpendent Public Health Consultant, Boston, Massachusetts, United States of America

* nathan@bu.edu

## Abstract

### Background

Methane leaks across the natural gas process chain, including in homes. To date, no studies have described how common it is to have elevated methane in homes served by gas, in comparison to homes without gas.

### Methods

In this study of homes in Massachusetts and Rhode Island, we utilized Cavity Ring-down Spectrometry to measure methane concentrations in outdoor air, and at mid-floor-level indoor air in basements and first, second, and third floors. We recruited a total of 195 homes in urban and rural areas, 175 of which had gas service and 20 of which did not.

### Results

Indoor $[CH_4]$ in households with gas service was elevated over outdoor $[CH_4]$, averaging 1.45 parts per million (ppm) elevation over outdoor ambient $[CH_4]$ (p < 0.0001– 0.0068), and up to 38.2 ppm above outdoor ambient $[CH_4]$. Ninety-three percent of homes with gas showed higher median indoor $[CH_4]$ than the median $[CH_4]$ in non-gas homes. By contrast, indoor $[CH_4]$ in gas-free homes did not differ from outdoor conditions (p > 0.10), except marginally on the first floor (0.10 ppm elevation; p = 0.04). In 91% of a subset of homes investigated, leaks from gas equipment were confirmed.

### Conclusions

Elevated $[CH_4]$ is common in homes served with gas. Gas leaks and incomplete combustion were identified as sources of elevated $[CH_4]$. There was no relationship between indoor $[CH_4]$ and home age or square footage; residents shouldn't assume that newer homes are less prone to indoor gas leaks. The majority of gas in the United States comes from hydraulically fractured gas containing carcinogenic

**Data availability statement:** Household data may be made available upon request by contacting the Institutional Review Board of Boston University, at irb@bu.edu.

**Funding:** This research was funded by Zero Carbon Massachusetts.

**Competing interests:** The others have declared that no competing interests exist.

co-pollutants. It is not well understood how consistent low-dose exposure to gas co-pollutants like mercaptans and benzene affects health. Additional studies could clarify any differences in health outcomes for people living in homes serviced by gas and those who don't use gas.

## Introduction

Methane leaks across the natural gas process chain create explosion risks [1], air pollution [2,3], climate damage [4], tree damage [5] and ratepayer expense [6,7]. The global monthly mean methane concentration ([$CH_4$]) is currently 1.94 parts per million, approaching a tripling of its pre-industrial level and having risen over the last decade at a rate comparable to the highest rates observed since global monitoring began in 1984 [8]. Methane is a powerful but short-lived greenhouse gas [9], which makes reducing methane leaks from gas infrastructure timely in addressing climate change. While indoor gas leaks have been studied for decades [10], emphasis on gas leaks in the household is a renewed focal point of health, safety, cost, and climate research. In addition to increasing evidence of negative climate impacts from incompletely combusted gas from appliances, [11,12] and health impacts from combusted gas in homes [13–15], recent studies have progressed from documenting health-concerning compounds in uncombusted gas piped into households [16,17], to documenting leaks of uncombusted gas from appliances [18–20], to the presence or emissions rate of gas compounds in household air [21–23], consistent with indoor gas leaks.

In this study we examine how commonplace elevated methane concentrations are in indoor ambient air in residential buildings across housing types in two New England states. We compare homes with gas service to those without gas. To conduct this survey, we took in-home measurements of indoor ambient air as an indicator of indoor gas leaks and incomplete combustion. We measured indoor methane with high precision in 195 different households. We obtained data from homes in a total of 52 municipalities (or designated neighborhoods of Boston), with 45 municipalities visited in Massachusetts and seven in Rhode Island. More specific information on homes are listed in the supplementary data document.

To our knowledge, this is the largest observational survey to examine detectable household methane elevations below 10 ppm to date. This study complements recent work by Rowland et al. [17]. In that study, researchers tested methane in indoor air across 323 homes. However, in the leak survey performed by those authors, the goal was "to emulate a general leak detection survey for near-Lower Explosive Limit (LEL) concentrations that would be performed by licensed appliance technicians (e.g., gas utility employees) or firefighters responding to a reported leak." The Lower Explosive Limit concentration of methane in air at standard temperature and pressure is between 44,000 and 50,000 parts per million or 5% of air volume [24]. The minimum elevation in [$CH_4$] used in Rowland, et al. [17] to designate a leak was 10 ppm above background, which would potentially miss a large number of homes with [$CH_4$] elevations below a 10 ppm threshold.

There are at least two key reasons why documenting the presence of elevated methane in indoor ambient room air less than 10 ppm is important. First, relatively large gas leaks that unnecessarily add to ratepayer bills and to climate warming may produce relatively smaller elevations of [CH$_4$] in homes with poor insulation and high air exchange rates compared to the same sized leak in better insulated and/or less well-ventilated homes [23]. Second, degree of exposure and potential health impacts of exposure to co-pollutants in low-level but continuous gas leaks are under-studied, including health impacts from benzene and other volatile organic compounds identified in piped residential gas, and added odorants [25]. Our study fills an important gap in determining the prevalence of elevated [CH$_4$] above outdoor concentrations in homes across a range of size and age, with or without gas service, that have not previously been examined.

## Materials and Methods

### Home selection

Homes in urban and rural areas were selected across eastern Massachusetts and Rhode Island using a combination of quota sampling [26] and snowball sampling [27]. Quota sampling was used to achieve a relatively balanced set of households of different types and ages (single-family, multi-family, and multi-unit apartments). Snowball sampling was used to recruit households via word of mouth and social networks. We recruited a total of 195 homes for this study, 175 of which had gas service and 20 of which had no gas service. Household addresses, including addresses provided by the funders of this research, were not shared beyond the investigators. A limitation of this study is that of the 195 households, only two were rentals. Indoor [CH$_4$] data from homes was collected on 40 dates during the period of February 24, 2024 to June 20, 2024.

Participants were asked to keep doors and windows closed prior to the visit, typically a day prior to visits, but normal traffic in and out of the homes was not curtailed. Participants were not asked to limit the use of gas cooking stoves or ensure that heating was on prior to arrival, as the study was meant to capture a moment of typical gas use inside a home. This study did not collect information on whether residents or researchers could smell gas upon our arrival to the home.

Measurements were taken during the day from 7am to 5 pm. Outdoor [CH$_4$] was taken just prior to entering the home with a stable reading for 5 seconds. Indoors, measurements were taken at approximately the middle of the floor level. Indoor and outdoor temperature were not tracked. No homes had repeated visits.

Methane concentration data collection:

We utilized Cavity Ringdown Spectrometry (Gas Scouter G4301 Mobile Gas Concentration Analyzer, Picarro, Inc., Santa Clara, CA USA) to visit homes to measure indoor air samples, in basements and first, second, and third floors as applicable/available. This instrument, with a raw precision of 0.003 ppm [CH$_4$], is suitable for detecting elevated [CH$_4$] in ambient room air at greater precision than the 1 ppm resolution instruments used in Rowland, et al. [17] for the ambient room air leak survey part of that study. Methane measurements were made in the center of rooms, without regard to proximity to gas appliances. Checks on the analyzer calibration were performed three times during the course of this study; near when sampling began (February 29, 2024), approximately midway during sampling (April 23, 2024), and near the conclusion of home sampling (June 19, 2024). Calibration check results are presented in Appendix 1.

Because outdoor ambient [CH$_4$] values can vary by up to hundreds of parts per billion on a diurnal basis, we sampled outside air immediately prior to entering, and immediately after exiting test homes. With a rising (10% −90%) or falling (90% − 10%) instrument response time of 5 seconds, we waited at least 30 seconds for readings to stabilize in an indoor or outdoor environment before recording values. We recorded the difference between indoor and outdoor methane so that each home studied had its own outdoor [CH$_4$] comparison value. This work was completed during late winter to late spring 2024, when New England homes typically keep windows and exterior-facing doors closed. To ensure comparable indoor conditions, householders were asked to minimize ventilation through windows or doors for at least a day prior to measurements. We did not request any further special weather sealing or insulation.

When possible and convenient for the householder, we conducted brief searches for sources of elevated [CH$_4$] inside homes, typically on pipes, pipe connections and gas appliances including stoves, water heaters and gas boilers. A combination of close-up [CH$_4$] readings and bubble observations from a soap solution applied to suspected leak points were used to pinpoint leaks that could contribute to elevated [CH$_4$] in the ambient room air.

### Data Analysis

Using RStudio© Version 2024.12.1+563, the Mann-Whitney U test was used to evaluate the comparison between no gas homes (n=20) and gas homes (n=175) because of the small sample size of no gas homes and the long tail distribution of leaks in the gas homes. We used the Wilcoxon Sign-Rank test for paired samples of outdoor and indoor air for no gas homes and a paired t-test was used to compare mean indoor and outdoor [CH$_4$] concentrations in the gas group. Linear regression was used to evaluate the association between elevated methane and home age and size.

## Results

### Housing characteristics

A breakdown of the housing types studied was single-family 63.5%, multi-family (2–3 units) 26.6%, and multi-family (4+units) 9.9%. There were two row houses and we categorized them as multi-family (2–3 units) because they shared a wall with another dwelling. For context, the US Census Bureau estimates that in 2023, 57.2% of Massachusetts and 61.1% of Rhode Island households are single-family. Natural gas is used for heating and cooking in about half the homes in Massachusetts [28] and Rhode Island [29]. In this study, among gas service households, 87% reported using gas stoves, 80% gas water heating, 75% gas space heating, 39% gas dryers, and 2% gas fireplaces.

The largest [CH$_4$] elevations were in basements, where [CH$_4$] concentrations were nearly double outdoor concentrations, with a mean 1.91 ppm above outdoor [CH$_4$], while [CH$_4$] in 1st, 2nd, and 3rd floors averaged about 1.29 ppm higher than outdoor ambient. In the gas group, mean indoor [CH$_4$] readings were higher than mean outdoor concentrations for all floors. By contrast, homes without gas service were either not significantly different from outdoor ambient (p>0.10, Table 1), or only marginally on first floors with a [CH$_4$] elevation of 0.10 ppm (p=0.04). These [CH$_4$] elevations in homes without gas service may have been caused by biotic factors such as human metabolism or improperly functioning plumbing.

Table 1. Summary statistics of [CH$_4$] concentrations in parts per million (ppm) outdoors and indoors. Indoor concentrations are unadjusted for outdoor concentrations. P-values are for differences between indoor and outdoor [CH$_4$].

| Category | Outdoor/ Indoor Floor | Mean (ppm) | Standard Deviation | Number of homes (n) | Standard Error | Median (ppm) | P-value |
|---|---|---|---|---|---|---|---|
| Gas | Outdoors | 2.04 | 0.10 | 175 | 0.001 | 2.02 | |
| No Gas | | 2.03 | 0.09 | 20 | 0.004 | 2.01 | |
| Gas | Basement | 3.95 | 3.80 | 164 | 0.023 | 2.83 | < 0.0001 |
| No Gas | | 2.06 | 0.27 | 18 | 0.015 | 2.02 | >0.10 |
| Gas | First Floor | 3.35 | 2.31 | 166* | 0.014 | 2.63 | < 0.0001 |
| No Gas | | 2.13 | 0.25 | 20 | 0.013 | 2.07 | 0.04 |
| Gas | 2nd Floor | 3.33 | 2.18 | 141 | 0.015 | 2.65 | < 0.0001 |
| No Gas | | 2.11 | 0.16 | 12 | 0.014 | 2.08 | >.10 |
| Gas | 3rd floor | 3.32 | 2.77 | 41 | 0.068 | 2.50 | 0.005 |
| No Gas | | 2.10 | 0.21 | 4 | 0.051 | 2.03 | ** |

*Note that the first floor of a dwelling unit may not be the first floor of a building, for example in multi-story apartments.

**P-value is not presented due to the small number of homes with 3rd floors in the no gas group.

In 195 homes studied, homes supplied with gas showed significantly higher median [CH₄] concentrations than homes not served by gas after outdoor methane concentrations were subtracted (Table 2, Fig 1).

The distribution of leak sizes in homes with gas service was not bell-shaped, as indicated by the downward shift of the median values of [CH₄] difference shown in Fig 1 (shown by the line in the box) from the mean [CH₄] values (shown with an X). As across other sectors of the fracked gas process chain [30,31], the distribution of leak sizes in homes with gas service were skewed with a "long tail" containing a small proportion of large [CH₄] values and a large proportion of homes with lower [CH₄] elevations (Fig 2).

Ninety-three percent of gas homes had methane concentrations higher than no gas homes and nearly half of gas homes had elevated methane concentrations greater than 1.0 ppm [CH₄] concentrations above outdoor [CH₄] levels.

We had the opportunity to conduct brief searches for suspected methane sources for elevated [CH₄] in 137 (78%) of the 175 homes with gas service. Of these homes, we positively identified leak sources in 124 homes, or in 91% of homes checked. Leak sources were found all along the distribution system within homes: pipe connection (63), appliance malfunction or incomplete burning (53), shutoff valves (13), migration (3) and corrosion (1). Eighteen (18) of the leaks came from gas company pipes. Beyond the statistical association of higher [CH₄] in homes with gas, this provides

Table 2. Summary statistics adjusted for outdoor [CH₄] concentrations. Indoor minus outdoor methane concentrations ([CH₄], parts per million) from 195 households surveyed in this study. P-values are for differences between gas and no gas homes.

| Category | Floor | Mean (ppm) | Standard Deviation | Number of homes (n) | Standard Error | Median (ppm) | P value |
|---|---|---|---|---|---|---|---|
| Gas | Basement | 1.92 | 3.81 | 164 | 0.30 | 0.80 | < 0.0001 |
| No Gas | | 0.03 | 0.18 | 18 | 0.04 | −0.01 | |
| Gas | 1st Floor | 1.32 | 2.30 | 166 | 0.18 | 0.6 | < 0.0001 |
| No Gas | | 0.10 | 0.19 | 20 | 0.04 | 0.03 | |
| Gas | 2nd Floor | 1.29 | 2.17 | 141 | 0.18 | 0.6 | < 0.0001 |
| No Gas | | 0.09 | 0.17 | 12 | 0.05 | 0.03 | |
| Gas | 3rd Floor | 1.20 | 2.80 | 41 | 0.48 | 0.49 | ** |
| No Gas | | 0.06 | 0.19 | 4 | 0.10 | −0.02 | |

*Note that the first floor of a dwelling unit may not be the first floor of a building, for example in multi-story apartments.

**P-value is not presented due to the small number of homes with 3rd floors in the no gas group.

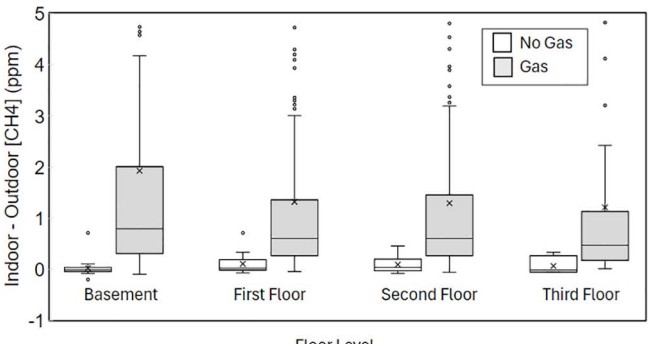

Fig 1. Adjusted indoor [CH₄] concentrations of gas and no gas households are shown by floor (basements, 1st, 2nd, and 3rd floors, from left to right). Not all homes had basements, 1st, or 2nd, and 3rd floors. The mean is denoted with an "X" and the median is shown with a line within the box. Outliers not shown extend up to 38 ppm. See Fig 2 for more information about outliers. "Gas" refers to households with gas service; "No Gas" refers to households without gas service, all-electric and without propane.

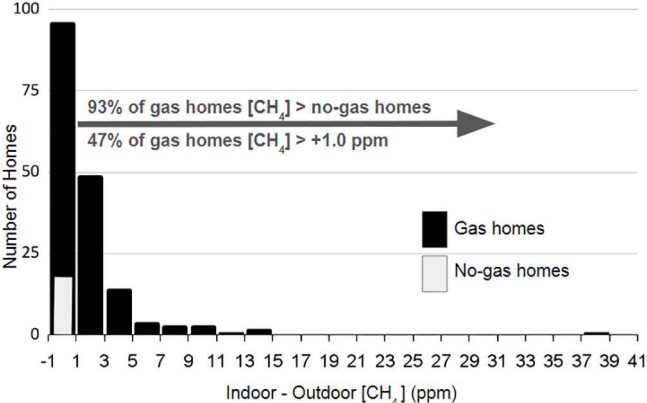

**Fig 2. Histogram showing numbers of homes with different levels of indoor minus outdoor methane ([CH₄]) concentrations.** [CH₄] shown is the highest [CH₄] obtained at any floor. Black bars represent homes with gas service; the gray bar represents homes without gas service. The + 1.0 ppm enhancement noted in the figure is equivalent to a 3.0 ppm total [CH₄] reading.

further evidence linking elevated [CH₄] in homes to incomplete combustion, and leaks from gas pipes, connections, and appliances.

There did not appear to be a relationship between home attributes or geography and elevated [CH₄] in homes served by gas. For example, the 10 homes showing the highest elevation of [CH₄] relative to outdoors were located in nine different municipalities and in both states. Of these top 10 homes, all contained basements and first floors, eight had second floors, and three had third floors, and all ten were found in single family homes. There was no relationship between indoor [CH₄] and either home age ($P > 0.1$) or square footage ($P > 0.1$). Thus, living in a relatively newer home should not be a reason to assume it is less likely to contain elevated methane. This study only included two rental units, and we weren't able to evaluate if there were differences between methane concentrations in owner-occupied homes and rental homes.

### Degraded service lines and safety

Although it was not an objective of this study to investigate leak origins or the condition of pipes, fittings and appliances, when possible, attribution of elevated [CH₄] to gas leaks was evaluated by identifying leak locations and incomplete combustion. In doing this work, one unanticipated finding was that three of 18 (17%) bare steel service lines (observed in the opportunistic process of leak origin determination) in homes served by gas were found to be in poor condition and called in for repair. The service line material can often be determined at the outside meter coupling or at the service entrance inside the basement of the home.

## Discussion

### Prevalence of elevated methane

We observed that in 175 homes served with gas, elevated indoor methane is not only common, but to be expected. Conversely, in gas-free homes, elevated indoor methane levels were found only marginally on first floors, with less than 1/10th the elevation of [CH₄] on average than in homes served with gas. Basements in gas homes had the highest methane concentrations compared to no gas homes and other floors in gas homes. In New England, basements are commonplace and are a typical place to locate a gas service line entrance, heating system, water heater, clothes dryer, and gas lines leading to other appliances. Incomplete combustion from basement appliances and gas leaks from service lines contributed to these higher concentrations.

Data collected in this study spanned a period from winter through late spring. Although we attempted to ensure comparable indoor conditions by requesting householders minimize ventilation at least a day prior to sampling, it is plausible that weather conditions and/or incomplete combustion in indoor gas appliances could have produced seasonally varying levels of indoor $[CH_4]$ enhancement. Yet in general we found that when there was an elevated indoor methane reading, there was a corresponding gas leak. We found gas stoves and water heaters had fitting leaks indoors even when off. Therefore, in our opinion having the heat on or off made little difference in our ability to detect elevated methane from a leak, although it is possible that whether the appliance was on or not may contributed to the magnitude of the leak.

Prior to this research, the only categorical finding on prevalence of detectable elevated $[CH_4]$ in indoor ambient air in gas-served homes reported 13 of 323 homes with a gas leak, or about 4% of households [17]. With the finer resolution analyzer used in this study, we estimate from the data presented in Fig 2 that 93% of homes served with gas had greater $[CH_4]$ than homes without gas. Reconciling these vastly different estimates is obtained by comparing the number of homes found with $[CH_4]$ of 10 ppm elevation relative to outdoors. In this study we find six homes of 175 gas-served homes, or 3.4% to compare with the 4% found in Rowland, et al. [17]. Analogous to the "super-emitter" character of gas leaks across the gas process chain, these findings indicate there could be gas leaks that are under explosive limits but go undetected by residents.

## Implications for safety and methane emissions

Our preliminary finding that one sixth of steel service lines were in poor condition is a safety concern and warrants focused attention on bare steel pipe condition. The majority of leak-prone service lines are bare steel [1]. We recommend that bare steel and other leak-prone service lines be inspected on an annual basis. There is currently no standard minimum frequency of required service line inspection across states.

A practical implication of our finding of a skewed distribution of $[CH_4]$ in homes is that homes with the largest $[CH_4]$ can be prioritized by regulators and utilities for repair, and leak flux rates can be assessed according to a recently developed method [23]. It is important to note that just because a home exhibits relatively large $[CH_4]$ does not mean it has a large flux rate, as it depends on the air exchange rate in the home. A small leak in a home with a tight envelope and low air exchange could show a large elevation in methane, while conversely a large leak in a home with a high level of air exchange could show a small elevation in methane.

This study was not designed to determine methane emission rates but rather stable concentrations, where steady indoor air composition is promoted by weather conditions where windows and doors are likely to be closed. To obtain rate estimates, at least two measurements must be made over a recorded time interval, with an additional informational constraint such as a known $[CH_4]$ starting point, or a known room or home air exchange rate. This approach to estimate emissions was made in Nicholas et al. [23]. That study, while having a distinctly different objective, nevertheless is in broad consistency with findings here. For example, Nicholas et al. [23] identified leaks in 18 of 20 (90%) of homes, similar to the 93% of homes with gas documented here that had $[CH_4]$ higher than the average of non-gas homes; and among the 137 homes in this study where we had permission to search for the methane source, we pinpointed the leak or incomplete combustion origin 90% of the time. Secondly, the suggestion of a long-tailed distribution with potential "super-emitters" in Nicholas et al. [23] is further supported by the long-tailed distribution found here (Fig 2) with almost ten times as many homes studied.

## Implications for air quality and health

These undetected methane concentrations in indoor air have health implications because of the potential for exposure to co-pollutants. Heating and cooking with natural gas is common in the Northeast, with half of households using gas in Massachusetts and Rhode Island. The majority of natural gas in the United States is extracted through a process called hydraulic fracturing and is commonly known as "fracked" gas. In the United States, 89% of U.S. dry natural gas production in 2022 was extracted using fracking, and its production is expected to continue to grow, with estimates of it making

up 93% of production by 2050 [32]. The presence of volatile organic compounds like the carcinogen benzene, have been characterized as part of the fracking process [33], found near fracking sites [34,35], and characterized in distributed gas used in residential end-uses in Massachusetts [16]. We distinguish "fracked" natural gas from conventional methods because of its significant growth in use since 2005 [32,36], known deleterious impact on water sources at the site of drilling [37], and because carcinogens used in the hydraulic fracturing process can affect human health at the well-sites, and potentially impact human health in the home where gas is used. Massachusetts gas contains carcinogenic compounds including benzene [16], of concern for human health due to both acute and chronic exposure [38,39].

In addition to carcinogens like benzene, odorants (mercaptans) in gas also have health implications. Mercaptans were documented to have acute health impacts, such as from a large gas leak in 2015 in Aliso Canyon, California [40]. While little is known about long-term, low-dose exposure to fracked gas odorants in residential settings, mercaptans are known to impact human health; chronic mercaptan exposure in occupational settings is regulated by the National Institute for Occupational Safety and Health (NIOSH) (REL C 0.5 ppm (1 mg/m3) [15-minute]) [41], and mercaptan exposure caused lung inflammation and apoptosis in rats [42].

While it is assumed people will be able to smell gas indoors, evidence suggests this assumption may not always be true. There are differences in concentration of odorants across the gas system [17], and differences in individual's ability to smell gas, may compromise the safety of gas in homes. People with poor or compromised sense of smell may be unaware of leaking gas. Compromised sense of smell is often not detected by individuals until it is advanced and is more common among the elderly [43]. People can also become adapted or habituated to persistent odors to the point that they no longer detect them [44]. On streets and sidewalks, our previous experience (e.g., in informal observations during research published [6]) is that many people, if focused on the task, can smell gas leaks at methane levels only 1 part per million (ppm) above the ambient background of 2 ppm, or at 3 ppm). In our study, 47% of households with gas that we studied had methane concentrations at or above 3 ppm. While they are detectable, at least to the occasional visitor with a keen nose, they may not be noticed by habituated or adapted residents. We recommend that gas companies be required to use equipment with a detection threshold of 1 ppm [$CH_4$] above ambient, and to be required to report elevations above 1 ppm [$CH_4$], relative to outdoors, in interior spaces to residents and in annual reports to state regulators.

Almost all relevant studies on health impacts of gas in homes to date have focused on health impacts of *combusted* (not leaked) fracked gas. Studies that have looked at the health impacts of unburned fracked gas focused predominantly on the upstream gas sector, near locations of fracking [45–47]. This study complements studies that document constituents of gas found in homes (e.g., Michanowicz, et al. [16], Lebel, et al. [22]) because it describes the high prevalence of elevated methane in residential homes. This study represents, to our knowledge, the largest household survey in New England using a comparison group and documenting the presence and prevalence of elevated methane in ambient room air and demonstrating its association with gas equipment in homes.

## Limitations and future research

While this study contributes knowledge that elevated [$CH_4$] in New England homes served by fracked gas is common, there are a number of study limitations: we did not measure fluxes of methane, concentrations of co-pollutants such as benzene, nor assess odorant detectability by residents. We did not sample comprehensively nor randomly across all housing stock and household types (including renters), household ventilation types or degree of insulation, and did not undertake a comprehensive survey of service line conditions. These limitations raise further questions for follow-on studies that are conducted in as large or larger sample sizes than the 195 homes studied here.

## Conclusions

Elevated methane is common in homes served with gas and associated with gas leaks and incomplete combustion. There was no relationship between indoor [$CH_4$] and home age or square footage; residents shouldn't assume that newer homes

 

are less prone to indoor gas leaks or incomplete combustion. The majority of gas in the United States comes from hydraulically fractured gas and has co-pollutants which are carcinogenic. It is not well understood how consistent low-dose exposure to fracked gas co-pollutants like benzene affect health. Additional studies could clarify if there are differences in benzene exposure and health outcomes for people living in homes serviced by gas and those who don't use gas.

## Supporting information

**S1 Appendix. Instrument calibration check.** We tested the analyzer prior to the beginning of the survey, on February 29, 2024; during a midpoint of the survey, on April 23, 2024, and near the conclusion of the survey on June 19, 2024, against nominal 0.0 ppm; 2.0 ppm and 10 ppm test gasses in ultrapure air. The test gas tanks (Scott-Marrin, Riverside, CA USA) were certified to contain < 0.01 ppm; 2.072 ppm; and 10.32 ppm [CH4] respectively (+/- 1% NIST). Test gasses were supplied to the analyzer using Tedlar bags filled with test gasses and connected to the analyzer inlet during normal operation. Tedlar bags of test gas supply were maintained until the graphical data display indicated the analyzer [CH4] reading stabilized near the nominal value. Table 3 shows the match between analyzer values of [CH4] and test gas values. These results demonstrate that our analyzer was working properly and with adequate precision for the study. (DOCX)

## Acknowledgments

We thank residents who volunteered their homes for this study, and Lisa Cunningham, David Mendels, and Kelly Salvatore for assisting in household recruitment.

## Author contributions

**Conceptualization:** Nathan G. Phillips, Robert Ackley, Andee Krasner.

**Data curation:** Nathan G. Phillips, Andee Krasner.

**Formal analysis:** Nathan G. Phillips, Andee Krasner.

**Funding acquisition:** Nathan G. Phillips, Robert Ackley.

**Investigation:** Robert Ackley.

**Methodology:** Nathan G. Phillips, Robert Ackley, Andee Krasner.

**Project administration:** Nathan G. Phillips, Robert Ackley, Andee Krasner.

**Supervision:** Nathan G. Phillips, Robert Ackley, Andee Krasner.

**Validation:** Nathan G. Phillips, Robert Ackley, Andee Krasner.

**Writing – original draft:** Nathan G. Phillips, Andee Krasner.

**Writing – review & editing:** Robert Ackley, Andee Krasner.

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
