## [Decision Letter · Decision Letter 0]

19 Dec 2025

PONE-D-25-25933Indoor methane consistently above outdoor levels in homes with natural gas servicePLOS One

Dear Dr. Phillips,

Thank you for submitting your manuscript to PLOS ONE. After careful consideration, we feel that it has merit but does not fully meet PLOS ONE’s publication criteria as it currently stands. Therefore, we invite you to submit a revised version of the manuscript that addresses the points raised during the review process.

We look forward to receiving your revised manuscript.

Kind regards,

Tharaga Sharmilan, Ph.D.

Academic Editor

PLOS One

Journal Requirements:

“Funding for this study was provided by ZeroCarbonMA.”

“This research was funded by Zero Carbon Massachusetts.”

“This research was funded by Zero Carbon Massachusetts.”

“The others have declared that no competing interests exist.”

We note that one or more of the authors are employed by a commercial company: “Gas Safety, Inc.”

5. We note that Figure 1 in your submission contain map images which may be copyrighted. All PLOS content is published under the Creative Commons Attribution License (CC BY 4.0), which means that the manuscript, images, and Supporting Information files will be freely available online, and any third party is permitted to access, download, copy, distribute, and use these materials in any way, even commercially, with proper attribution. For these reasons, we cannot publish previously copyrighted maps or satellite images created using proprietary data, such as Google software (Google Maps, Street View, and Earth). For more information, see our copyright guidelines: http://journals.plos.org/plosone/s/licenses-and-copyright.

1. You may seek permission from the original copyright holder of Figure(s) [#] to publish the content specifically under the CC BY 4.0 license.

6. Please upload a copy of Figure 4, to which you refer in your text on page 11. If the figure is no longer to be included as part of the submission please remove all reference to it within the text.

7. We note that there is identifying data in the Supporting Information file “PhillipsAckleyKrasner2025SupportingInformation.xlsx”. Due to the inclusion of these potentially identifying data, we have removed this file from your file inventory. Prior to sharing human research participant data, authors should consult with an ethics committee to ensure data are shared in accordance with participant consent and all applicable local laws.

-Location data

Please remove or anonymize all personal information “Address”, ensure that the data shared are in accordance with participant consent, and re-upload a fully anonymized data set. Please note that spreadsheet columns with personal information must be removed and not hidden as all hidden columns will appear in the published file.

Reviewers' comments:

Reviewer's Responses to Questions

**Comments to the Author**

1. Is the manuscript technically sound, and do the data support the conclusions?

Reviewer #1: Yes

Reviewer #2: Partly

2. Has the statistical analysis been performed appropriately and rigorously? 

Reviewer #1: I Don't Know

Reviewer #2: Yes

3. Have the authors made all data underlying the findings in their manuscript fully available?

Reviewer #1: Yes

Reviewer #2: Yes

4. Is the manuscript presented in an intelligible fashion and written in standard English?

Reviewer #1: Yes

Reviewer #2: No

5. Review Comments to the Author

Reviewer #1: The paper is very well written and I only have a few minor comments to address:

“relatively large gas leaks that 116 unnecessarily add to ratepayer bills and to climate warming may produce smaller elevations of 117 [CH4] due to poor home insulation and high air exchange rates” Smaller than what? It’s unclear what this is referencing.

Line 144: How long were residents instructed to keep their windows and doors closed before the visit? How long would be required to ensure the measurement is reflective of worst-case scenario?

Line 138: For the data collected in the warmer months, I’m not sure the heating being ‘on’ would have much an impact. Even though it might be ‘on’ on the thermostat it’s unlikely there would be calls for heating how was this addressed?

On line 52, it states that measurements were taken at floor level but on line 152, it says measurements were taken in the middle of the floor level. Which was it? Was there any indication of heating system runtime or gas appliance use during the measurement period? Also, how frequently were the measurements taken between 7am-5pm in each home.

On line 60/61, it says that leaks from gas equipment were confirmed but on line 178, it states this was done when convenient/possible for homeowner.

Line 209, what could cause the marginal methane increases in non-gas homes?

Reviewer #2: This manuscript examines a significant and urgent public health and environmental concern: the elevated indoor methane levels in homes utilising natural gas services. This subject pertains to indoor air quality research, climate science, and energy policy. The authors provide empirical measurements obtained from various households, offering valuable evidence of methane accumulation in real-world scenarios.

The study is well-motivated, and the results provide useful information that adds to the growing body of research on indoor emissions from natural gas appliances. There are, however, a few places where the manuscript could use more explanation, more methodological detail, and a stronger discussion of the context. These changes will make the results more scientifically sound and easier to understand.

Major Comments

1. Although the general sampling method is explained, more information is needed to make it possible to repeat the process. Please explain: • The reasons for choosing the homes that will participate (for example, the age of the building, the type of building, and the way it is ventilated).

• If sampling was done during certain times of the year or with controlled ventilation.

• The calibration and detection limits of the methane monitors used.

• If any co-pollutants (like NO₂, CO₂, or VOCs) were measured to put gas use or combustion events in context.

More information will make people more sure that they can compare samples from inside and outside.

2. The manuscript would benefit from further elucidation regarding:

• The identification and management of outliers.

• If indoor-outdoor differences were tested statistically (e.g., paired t-tests, Wilcoxon).

• How measurements that change over time (like cooking times and heating cycles) were dealt with.

It would be beneficial to clearly indicate whether the observed differences are statistically significant and how the variability among households was addressed.

3. There are a number of things that can affect indoor methane levels, like how much air flows through the house, whether there are gas leaks, how often appliances are used, and how tight the house is. However, these factors are not controlled in a systematic way. Please let us know if any leak detection or verification was done.

• The behaviour of the people living there or the time of year affected the measurements.

• Conditions outside (like wind and temperature) may have affected the gradients between inside and outside.

A brief discussion of sensitivity would bolster the conclusions.

4. The conversation brings up important issues about the effects of climate change and the quality of indoor air, but the implications could be made clearer. Things to think about:

• How this study compares to other recent studies on gas or methane leaks in buildings.

• The importance of the measured concentrations in relation to safety thresholds, even if they are much lower than acute hazards.

• Possible policies or ways to reduce the problem (for example, appliance maintenance or ventilation advice).

The manuscript would be better if it had a little more context.

Minor comments

1. Please make sure that all units (ppm, ppb) are used the same way in the text and figures.

2. Legends and axis labels on figures would be easier to understand, especially on graphs with more than one panel.

3. A table that lists the characteristics of the household (like having a gas stove or furnace, the age of the home, etc.) would help readers understand the differences.

4. Some of the sentences in the introduction sound the same; making them shorter will help the flow.

5. Please check for small grammar mistakes and tense inconsistencies.

6. PLOS authors have the option to publish the peer review history of their article (what does this mean?). If published, this will include your full peer review and any attached files.

Reviewer #1: No

Reviewer #2: No

---

## [Author Response · Author response to Decision Letter 1]

12 Feb 2026

We have responded in detail to each of the editor and reviewer comments, as an appendage to our cover letter.

---

## [Decision Letter · Decision Letter 1]

16 Apr 2026

Indoor methane consistently above outdoor levels in homes with natural gas service

PONE-D-25-25933R1

Dear Dr. Nathan Glen Phillips,

We’re pleased to inform you that your manuscript has been judged scientifically suitable for publication and will be formally accepted for publication once it meets all outstanding technical requirements.

Kind regards,

Tharaga Sharmilan, Ph.D.

Academic Editor

PLOS One

Additional Editor Comments (optional):

Reviewers' comments:

Reviewer's Responses to Questions

**Comments to the Author**

1. If the authors have adequately addressed your comments raised in a previous round of review and you feel that this manuscript is now acceptable for publication, you may indicate that here to bypass the “Comments to the Author” section, enter your conflict of interest statement in the “Confidential to Editor” section, and submit your "Accept" recommendation.

Reviewer #1: All comments have been addressed

Reviewer #2: All comments have been addressed

2. Is the manuscript technically sound, and do the data support the conclusions?

Reviewer #1: Yes

Reviewer #2: Yes

3. Has the statistical analysis been performed appropriately and rigorously? 

Reviewer #1: Yes

Reviewer #2: Yes

4. Have the authors made all data underlying the findings in their manuscript fully available?

Reviewer #1: No

Reviewer #2: Yes

5. Is the manuscript presented in an intelligible fashion and written in standard English?

Reviewer #1: Yes

Reviewer #2: Yes

6. Review Comments to the Author

Reviewer #1: The authors have addressed all my comments, thank you. I entered "no" for the data availability as that is what the authors had done on their form.

Reviewer #2: The manuscript addresses an important and timely topic, and its overall structure is clear and logically presented. However, the language requires refinement to meet PLOS ONE standards of clarity, precision, and readability. Although the manuscript is generally understandable, there are recurring issues with sentence construction, punctuation, and stylistic consistency. Several sentences are overly long or contain grammatical inconsistencies that affect clarity, particularly in the Introduction and Discussion sections. In addition, some informal phrasing and instances of repetition should be revised to achieve a more formal scientific tone.

Specific areas that need attention include:

Occasional awkward phrasing and run‑on sentences, especially in the Discussion section

Inconsistent terminology and formatting, such as the use of [CH₄] and the presentation of units

Minor typographical issues involving spacing, punctuation, and citation formatting

Sections where the language is overly assertive and would benefit from more cautious, evidence‑based phrasing

The abstract is generally clear, but it would benefit from improved conciseness and sharper focus, particularly in the concluding statements regarding broader implications.

Overall, the manuscript is coherent, but thorough language editing is recommended to ensure clarity, precision, and consistency throughout before publication.

7. PLOS authors have the option to publish the peer review history of their article (what does this mean?). If published, this will include your full peer review and any attached files.

Reviewer #1: No

Reviewer #2: No

---

## [Editor Report · Acceptance letter]

PONE-D-25-25933R1

PLOS One

Dear Dr. Phillips,

I'm pleased to inform you that your manuscript has been deemed suitable for publication in PLOS One. Congratulations! Your manuscript is now being handed over to our production team.

Kind regards,

on behalf of

Dr. Tharaga Sharmilan

Academic Editor

PLOS One